# Restaurant staff's knowledge, practices, and attitudes pertaining to food allergy in Qassim region, Saudi Arabia: A cross-sectional analysis

Ghadi A. Alkhalaf[1], Norah A. Aljuaylan[1]*, Jolan S. Alsaud[2], Boshra A. Aljokaidb[1], Fatima R. Aljalaood[1]

1 Patient Friends Association, Unaizah, Qassim, Saudi Arabia, 2 Family Medicine, Qassim Health Cluster, Qassim, Saudi Arbia

* noonaljailan@gmail.com

**Data Availability Statement:** All relevant data are within the manuscript.

## Abstract

Food allergies, often triggered by minute amounts of certain foods, affect people of all ages and frequently occur in dining-out settings. Previous research in Saudi Arabia has not investigated the extent of restaurant workers' knowledge of food allergies and their role in protecting patrons from allergic reactions. This descriptive study assessed the knowledge, attitudes, and practices of restaurant staff regarding food allergies in Qassim, Saudi Arabia between January and March 2023. Interviews and observations were conducted in four stages, and data were analyzed using the Environmental Health Specialists Network Food Allergen Study Protocol. Results showed that most restaurant staff had limited knowledge of food allergens and their symptoms, and a small proportion had received specific training on food allergies. Moreover, only 14% of restaurants provided allergen information on their menus. Despite these knowledge gaps, most staff exhibited positive attitudes toward managing food allergies. Findings also indicated that factors such as experience, training, and restaurant policies were associated with higher levels of knowledge and more positive attitudes. These results highlight the urgent need for the restaurant sector to implement food allergy measures, including clear policies and comprehensive training, to prevent potentially life-threatening incidents.

## Introduction

Food allergies (FAs) are immune-mediated responses to proteins that range from swift and occasionally life-threatening reactions to enduring ailments [1]. FAs affect individuals across diverse age groups and are not limited to those with compromised immune systems. Immunoglobulin E (IgE) mediates the immunologic reaction to dietary components [2]. Notably, recent global studies have reported an escalating prevalence of FAs, particularly IgE-mediated variants, affecting approximately 10% of children worldwide [3]. Furthermore, the World Allergy Organization reported that approximately 240 million individuals (3%) worldwide are affected by FAs [4]. These allergies and other dietary reactions significantly impact the lives of millions of Americans and their families [5].

**Funding:** The author(s) received no specific funding for this work.

A noteworthy statistic from the National Nutrition Committee at the Saudi Food and Drug Authority (SFDA), in collaboration with Taibah University, indicates that 21% of the Saudi population is affected by FAs, which manifest as symptoms ranging from mild to severe and potentially life-threatening. The weighted national prevalence rate for Saudi adults reporting at least one FA is 19.7%, with 16.5% experiencing a single FA and 3.9% experiencing multiple FAs [6]. Common dietary allergens that induce adverse responses in children and adults include milk, eggs, wheat, soybeans, peanuts, fish, tree nuts, sesame, and shellfish [4]. Even small food quantities can induce severe reactions, so stringent allergen avoidance measures and swift detection of and response to reactions are essential [4]. The manifestation and intensity of FA symptoms vary considerably among individuals and within the same individual over time [4,7]. Symptoms can manifest throughout the body, including the skin, gastrointestinal tract, and respiratory system [6]. Mucosal symptoms, such as watery eyes, might occur due to allergen exposure, and food-related itching is a skin condition commonly reported worldwide. Gastrointestinal symptoms include nausea and vomiting, whereas upper respiratory symptoms include nasal congestion. Lower respiratory issues manifest as breathing difficulties, and cardiovascular symptoms, such as elevated blood pressure, may be accompanied by mental or emotional manifestations, such as a sense of impending doom [8].

Current FA therapies remain controversial, lacking official approval from the Food and Drug Administration [9]. Existing approaches primarily advocate allergen avoidance, but there are challenges to avoiding common allergens often integrated into diets. In the case of exposure, emergency interventions, such as epinephrine administration, are recommended to address symptoms [10]. Therefore, the most important part of FA management is educating patients and their support networks on aspects such as cross-contamination, vigilant food label scrutiny, and rapid identification and management of allergic reactions [11].

The avoidance of allergenic foods in one's daily life poses a significant challenge, particularly when individuals have limited control over their diets, such as when dining out or in restaurants. In such instances, allergenic exposure often results from food contamination or the inclusion of ingredients beyond consumer expectations. Even when patrons communicate their dietary requirements to restaurant staff, the actual implementation of suitable measures depends on staff knowledge, attitudes, and subsequent actions [12]. Growing emphasis on fostering awareness of FAs among restaurant employees has emerged owing to indications of potential knowledge gaps identified in previous research.

Public health authorities have increasingly emphasized the need for food analysis knowledge and practices among food handlers and restaurant personnel, acknowledging their potential to mitigate the impact of FAs and enhance adherence to safety regulations. This is especially crucial for individuals such as children, teenagers, and older adults, who may not reliably identify potentially unsafe meals and rely heavily on foodservice staff or caregivers to mitigate risks [13]. In 2019, the SFDA mandated compliance with the Saudi Technical Regulation "SFDA.FD 56/2018," titled "Disclosure of allergens in the list of meals for food establishments that provide food to the consumer outside the home" [14].

Notably, several studies have sought to enhance restaurant personnel's understanding and practices regarding food analysis, encompassing nutritional, clinical, and food hygiene dimensions and safe farm-to-fork approaches to minimize the occurrence of FAs [12]. Furthermore, most studies have indicated that participants have a basic comprehension of metabolism [15]. In addition, research conducted in the United States has indicated that managers, food servers, and workers typically demonstrate positive attitudes and knowledge regarding FAs among customers. Although FA training does not correlate with knowledge levels among various groups, it is associated with managers' and servers' attitudes [16].

To the best of our knowledge, research in this field is limited in Saudi Arabia; no study has comprehensively examined restaurant staff's knowledge and attitudes regarding FAs. Therefore, this study gathered descriptive data on restaurant workers' knowledge, attitudes, and practices pertaining to FAs.

## Materials and methods

### Study design and setting

This study employed a descriptive cross-sectional design to investigate the knowledge, attitudes, and practices of restaurant staff regarding food allergies in Qassim, Saudi Arabia and followed the Environmental Health Specialists Network (EHS-Net) Food Allergen Study Protocol [17]. Data were collected from restaurants located in Unaizah, a city within the Qassim region.

### Study population and sampling

The study population consisted of restaurant managers, food workers, and servers in Unaizah. Restaurants were included if they prepared and served food or beverages to customers, excluding food carts, mobile food units, supermarkets, temporary food kiosks, catering services, or in-store restaurants.

### Data collection

Data were collected between January and March 2023 using a four-stage approach based on the Environmental Health Specialists Network (EHS-Net) Food Allergen Study Protocol [17]:

1. Restaurant Selection: A list of eligible restaurants was created based on the inclusion criteria.

2. Prior Approval: Prior approval for visits was obtained through telephone conversations with restaurant owners after explaining the purpose of the study, the participant's rights to confidentiality, and withdrawal at any time without any re-sponsibility towards the study team.

3. Informed Consent: Written informed consent was obtained from restaurant owners, managers, food workers, and servers.

4. Data Collection: Interviews and observations were conducted by members of the Scientific Research Unit (SRU) at the Patients' Friends Association in Unaizah:

- Manager Interviews: Interviews were conducted with restaurant managers (kitchen authority).

- Food Worker Interviews: Interviews were conducted with food workers primarily engaged in food preparation or cooking.

- Server Interviews: Interviews were conducted with servers responsible for taking orders and serving customers.

- Observations: Observations were also conducted to assess the availability of suitable resources and conditions for catering to individuals' food allergy needs. An SRU representative monitored this process.

- Participant Selection: Restaurant managers selected food workers and servers for solicitation to ensure active participation.

We asked nineteen questions during the interviews to assess FA knowledge among managers, food workers, and servers. These questions aimed to gauge proficiency in recognizing major food allergens and comprehending appropriate responses to adverse food allergic reactions. In addition, we employed a Likert scale to evaluate staff attitudes toward FA through five questions. For the attitude score, we assigned point values to each response as follows: strongly disagree = 1, disagree = 2, unsure = 3, agree = 4, and strongly agree = 5. We then averaged each participant's response to the five attitude questions. The questionnaire encompassed 13–22 queries regarding FA practices, covering topics such as the restaurant's intention to address queries from customers with FA and the availability of designated personnel to handle FA-related inquiries and requests.

Furthermore, the members collecting data observed the restaurant and menu, appraising further restaurant attributes (highest-priced food item) and scrutinizing FA-related documentation (menu allergen mentions and availability of allergen-related documentation in the kitchen area). These assessment standards aligned with EHS-Net guidelines, which aim to enhance comprehension of factors influencing food safety [16].

## Ethical considerations

This study obtained ethical approval from the Regional Research Ethics Committee, Qasim Province (NCBE, No. H-04-Q-001) for the implementation and publication of the study (registration number 607-44-5119, approval date: October 31, 2022). Informed consent explaining the purpose of the study, the participant's rights to confidentiality and withdrawal at any time without any responsibility towards the study team, and how the data will be managed was obtained from restaurants' owners, managers, food workers, and servers.

## Data analysis

Data were analyzed in two stages: 1) Data Cleaning and Editing: Data were cleaned, edited, and recoded to ensure accuracy and consistency. A frequency analysis was performed for each variable to identify instances of non-response and bizarre replies. Variables with a large number of non-responses or poor quality were excluded.2) Descriptive Analysis: Descriptive statistics, such as frequencies and percentages, were used to summarize and describe the data. Cross-tabulations and univariate frequencies were also conducted for the selected demographic variables (independent vs. chain restaurants).

Statistical Package for the Social Sciences (SPSS version 23) was used for data entry, modification, and analysis. Specifically, descriptive statistics such as frequencies and percentages were employed to summarize and describe the data, providing a clear understanding of the distribution and characteristics of the variables under investigation. Additionally, we used analysis through the total analysis of variance (ANOVA) to assess the statistical significance of differences between the means of different groups. Statistical significance was considered present if $p < 0.05$.

Microsoft Excel (version 2021) was utilized to create visually appealing charts and graphs to enhance the presentation of findings. These graphical representations facilitated the communication of results and made it easier for stakeholders to grasp the key trends and patterns within the data.

## Results

### Restaurants' attributes

Among the 328 restaurants that were approached to participate in this study, 250 were eligible, and 178 (71%) agreed to participate. Data from the managers' interviews indicated that 72% of

the participating restaurants were independently owned. Furthermore, 48% of the restaurants visited did not receive any critical violations in the last inspection, 2% received one critical violation, and 1% received three violations. Regarding the languages that workers spoke most often while at work, Arabic ranked first (88%). In addition, 32% of restaurants provided >100 meals daily. The members collecting data classified 56% of the visited restaurants in Unaizah as providing full-service casual dining (Table 1).

## Manager, food worker, and server characteristics

Interview data from the 178 managers indicated that 97% were men, 88% of the participants primarily used Arabic as their language of communication, 69% had a high school diploma or lower, at least 73% had been employed in a restaurant for a minimum of two years, and 17% had obtained a food safety certification. Furthermore, 11% of the managers had received FA training undertaken while employed at their restaurant, and 38% had not served any meals to customers with FAs during the previous month (Table 1).

The interview data collected from 178 food workers revealed that 97% were men, 78% had a high school diploma or less, and 87% spoke Arabic. Approximately 73% of the restaurant workers had been working at the restaurant for ≤2 years, 92% had not received FA training while employed at their restaurant, and 22% did not recall meal preparation for customers with FAs in the preceding month (Table 1).

Interview data from the 178 servers revealed that 99% were men, 85% used Arabic as their primary language, and 86% had a high school diploma or less. Approximately 72% of the restaurant servers had been working in the restaurant for ≤2 years. While working at their current restaurant, 5% had received food safety training, and 39% did not remember serving meals to customers with FAs in the past month (Table 1).

## Observations

Notably, a significant portion (86%) of the surveyed restaurants did not provide any allergen-related information in their food list, whereas the remaining 14% did. In addition, 91% of the restaurants lacked allergen documentation in the front-of-house or dining areas. Among the establishments with such documentation, the signs encompassed a comprehensive list of allergenic ingredients, including cereals, fish, peanuts, nuts, milk, celery, eggs, mustard, sesame seeds, mollusks, legumes, soybeans, and wheat. A case in which a restaurant's sign included the directive "Please let us know when ordering if you are allergic" was observed. Furthermore, tableside placards were occasionally spotted, containing information about allergenic foods frequently featured on the menu (Table 2).

Findings further revealed that 95% of the establishments lacked any form of allergen documentation or informational panels in their kitchen areas. Among the restaurants, refrigerators exhibited varying conditions, with some proving to be well-maintained, hygienic, and secure. Conversely, other restaurants lacked an allergy list, and certain establishments restricted customer access to refrigerated items. Notably, Arabic was the exclusive language for documenting allergen-related information across all restaurants displaying signs or documents concerning allergens (Table 2).

## Manager, food server, and worker knowledge

Managers correctly identified peanuts (65%), milk and dairy products (59%), shellfish (33%), and eggs (71%) as major allergens and recognized difficulty breathing (55%), hives or rashes (81%), and swelling of the tongue and throat (57%) as symptoms of an allergic reaction to food. Almost all managers (80%) knew to call 937 when a customer had a severe allergic

**Table 1. Data describing the characteristics of the restaurants, managers, and staff.**

| Characteristics | n (%) |
|---|---|
| **Restaurant characteristics (a)** | |
| Restaurant type (n = 176) | |
| Independent | 126 (72) |
| Chain | 50 (28) |
| **Service type (n = 159) (b)** | |
| Fine or full-service casual dining | 89 (56) |
| Quick service or takeout-only | 70 (44) |
| Menu type (n = 178) | |
| Asian | 19 (11) |
| Non-Asian | 159 (89) |
| Number of meals served per day (n = 102) | |
| 1–50 | 38 (37) |
| 51–100 | 31 (31) |
| >100 | 33 (32) |
| Number of managers or individuals responsible for restaurant's operation (n = 157) | |
| ≤5 | 130 (83) |
| >5 | 27 (17) |
| Number of non-managerial restaurant employees (n = 163) | |
| ≤10 | 143 (88) |
| >10 | 20 (12) |
| The most expensive food item listed on the menu (n = 168) (b) | |
| ≤50 SR | 127 (76) |
| 51–100 SR | 14 (8) |
| >100 SR | 27 (16) |
| Number of severe violations reported at the most recent inspection (n = 178) b | |
| 0 | 85 (48) |
| 1 | 4 (2) |
| >1 | 1 (1) |
| Don't know | 88 (49) |
| **Characteristics of managers (a)** | |
| Sex (n = 177) | |
| Men | 172 (97) |
| Women | 5 (3) |
| Main language spoken (n = 178) | |
| Arabic | 156 (88) |
| English | 16 (9) |
| Other | 6 (3) |
| Highest educational attainment (n = 143) | |
| High school diploma or lower | 99 (69) |
| College level or higher | 44 (31) |
| Experience as a restaurant manager (n = 129) | |
| ≤2 years | 81 (63) |
| >2 years | 48 (37) |
| Ever obtained food safety certification (n = 162) | |
| No | 135 (83) |
| Yes | 27 (17) |
| Received food allergy training while employed at the restaurant (n = 165) | |

*(Continued)*

**Table 1.** (Continued)

| Characteristics | n (%) |
|---|---|
| Yes | 18 (11) |
| No | 147 (89) |
| Number of meals provided to customers with food allergies in the previous month (n = 95) | |
| 0 | 36 (38) |
| ≤10 | 43 (45) |
| >10 | 16 (17) |
| **Characteristics of foodservice workers (c)** | |
| Sex (n = 176) | |
| Men | 170 (97) |
| Women | 6 (3) |
| Main language spoken (n = 178) | |
| Arabic | 154 (87) |
| English | 18 (10) |
| Other | 6 (3) |
| Highest educational attainment (n = 122) | |
| High school diploma or lower | 95 (78) |
| College level or higher | 27 (22) |
| Years of experience as a restaurant service worker (n = 139) | |
| ≤2 | 101 (73) |
| >2 | 38 (27) |
| Received food allergy training while employed at the restaurant (n = 169) | |
| Yes | 13 (8) |
| No | 156 (92) |
| Number of meals provided to customers with food allergies in the previous month (n = 77) | |
| 0 | 17 (22) |
| 1–10 | 30 (39) |
| >10 | 30 (39) |
| **Characteristics of servers (d)** | |
| Sex (n = 178) | |
| Men | 176 (99) |
| Women | 2 (1) |
| Main language spoken (n = 178) | |
| Arabic | 151 (85) |
| English | 21 (12) |
| Other | 6 (3) |
| Highest educational attainment (n = 133) | |
| High school diploma or lower | 115 (86) |
| College level or higher | 18 (14) |
| Years of experience as a restaurant service server (n = 127) | |
| ≤2 | 92 (72) |
| >2 | 35 (28) |
| Received food allergy training while employed at the restaurant (n = 166) | |
| Yes | 8 (5) |
| No | 158 (95) |
| Number of meals provided to customers with food allergies in the previous month (n = 67) | |
| 0 | 26 (39) |
| 1–10 | 33 (49) |

(*Continued*)

**Table 1.** (Continued)

| Characteristics | n (%) |
|---|---|
| >10 | 8 (12) |

(a) Data from interviews with managers.

(b) Data from the observations of those collecting data.

(c) Data from interviews with food workers.

(d) Data from interviews with servers.

reaction to food, such as difficulty breathing. Managers (23%) knew that people who eat food to which they are allergic might die, and 47% of managers correctly said that removing food allergens from meals after preparing them does not safely prepare them for customers with FA.

Regarding restaurant workers' knowledge of the most common or major food allergens, we observed that 59%, 64%, 35%, and 71% of the workers believed that peanuts, milk or dairy products, shellfish, and eggs, respectively, were among the main allergens.

Furthermore, 17% of the workers were aware that a person with FA could die from eating a food they are allergic to, and 76% correctly said that taking a food allergen out of a meal after it

**Table 2. Descriptive information regarding food allergy protocols and observations of the restaurant environment.**

| Parameter | n (%) | |
|---|---|---|
| **Practices (a)** | | |
| The restaurant has a strategy in place to respond to inquiries from customers with food allergies (n = 169) | | |
| Yes | 30 (18) | |
| No | 139 (82) | |
| A designated individual is typically responsible for addressing questions and requests related to food allergies (n = 169) | | |
| Yes | 127 (75) | |
| No | 42 (25) | |
| **Observations (b)** | | |
| Does the menu display any information regarding allergens? (n = 174) | | |
| Yes | 25 (14) | |
| No | 149 (86) | |
| Information about allergens is accessible to customers in the front of the restaurant or dining area (n = 170) | | |
| Yes | 16 (9) | |
| No | 154 (91) | |
| Documents related to allergens are accessible within the kitchen area (n = 146) | | |
| Yes | 7 (5) | |
| No | 139 (95) | |

(a) Data were obtained from interviews with managers.

(b) Data were obtained from the observations of those collecting data.

has been prepared will not make it safe for a customer with FA. In addition, 57% and 61% of the workers correctly identified difficulty breathing and fever, respectively, as symptoms of an allergic reaction to food. Approximately 80% of the workers knew that they should call 937 for emergency services.

Regarding the restaurant servers' knowledge about the most common or major food allergens, we observed that 52%, 62%, 36%, and 70% of the servers identified that peanuts, milk or dairy products, shellfish, and eggs, respectively, were among the main allergens. Almost all servers (79%) knew to call 937 when a customer had a bad FA reaction, such as difficulty breathing. Servers (17%) knew that a person could die from eating food they are allergic to, and 75% of the servers correctly said that taking a food allergen out of a meal after the meal had been prepared would not make it safe for customers with FAs. Moreover, servers recognized difficulty breathing (50%), hives or rashes (76%), and swelling of the tongue and throat (55%) as symptoms of an allergic reaction to food (Table 3).

## Comparisons of manager, food worker, and server knowledge scores

Knowledge scores across the three distinct groups, as presented in Table 4, were significantly similar. Managers, food workers, and servers all exhibited a median knowledge score of 24 (mean = 22.7, standard deviation (SD) = 6.45, n = 211; mean = 23.3, SD = 6.51, n = 221; mean = 22.9, SD = 6.8, n = 197, respectively).

## Multiple logistic regression of the knowledge of managers, food workers, and servers

Multivariate logistic regression analysis revealed that two significant factors were associated with managers' enhanced FA knowledge scores (Table 5). Managers who had catered to >10 meals for allergic customers in the preceding month were more likely to possess superior FA knowledge over their counterparts serving ≤10 of such meals. In addition, restaurant managers equipped with specialists to address FA inquiries and requests exhibited elevated FA knowledge scores versus managers in establishments lacking such expertise.

Regarding food workers, multivariate logistic regression analysis revealed four significant factors associated with FA awareness (Table 5). Food workers in restaurants with established protocols for addressing queries from patrons with FA were more likely to have higher FA knowledge scores than those without such strategies. Female food workers were more likely to exhibit enhanced FA knowledge scores than their male counterparts. Food workers with at least two years of restaurant experience demonstrated heightened FA knowledge, surpassing their less-experienced counterparts. Notably, food workers stationed at restaurants where the most expensive food items ranged from 50 SR to 100 SR were more likely to possess superior FA knowledge scores over those in establishments offering items priced under 50 SR. Regarding servers, multivariate logistic regression analysis revealed three significant factors associated with enhanced FA knowledge scores. Servers assigned to address FA inquiries and requests in their establishments were more likely to exhibit elevated FA knowledge. Servers working in full-service restaurants were more likely to possess enhanced FA knowledge than their counterparts in quick-service establishments. Notably, servers employed in restaurants serving >300 meals daily were more likely to have superior FA knowledge over those serving ≤300 meals or fewer daily.

## Comparisons of manager, food worker, and server attitude scores

The median attitude scores for the three participant groups were approximately equivalent: 4.0 for managers (mean = 3.86, SD = 0.636, n = 211, r = 0.07, P = 0.371), 3.83 for food workers

**Table 3. Descriptive data on restaurant managers' and staff's food allergy knowledge (a).**

| | Manager (N = 178) | Food worker (N = 178) | Server (N = 178) |
|---|---|---|---|
| Question | N (%) | N (%) | N (%) |
| Which of the following do you consider to be significant allergens? | | | |
| Peanuts (correct) | 116 (65) | 105 (59) | 92 (52) |
| Milk or dairy products (correct) | 105 (59) | 113 (64) | 110 (62) |
| Tomatoes | 40 (23) | 45 (25) | 47 (26) |
| Shellfish (correct) | 59 (33) | 63 (35) | 64 (36) |
| Strawberries | 35 (20) | 33 (19) | 39 (22) |
| Chocolate | 52 (29) | 43 (24) | 45 (25) |
| Eggs (correct) | 126 (71) | 127 (71) | 124 (70) |
| Which of the following are symptoms of a food allergy? | | | |
| Difficulty breathing (correct) | 97 (55) | 102 (57) | 89 (50) |
| Hives or rash (correct) | 144 (81) | 142 (80) | 135 (76) |
| Headache | 52 (29) | 65 (37) | 61 (34) |
| Swelling of tongue and throat (correct) | 101 (57) | 97 (55) | 98 (55) |
| Fever | 99 (56) | 109 (61) | 109 (61) |
| What should you do if a customer experiences a severe allergic reaction, such as difficulty breathing? | | | |
| Recommend that they drink water | 89 (50) | 99 (56) | 106 (60) |
| Call 937 (correct) | 142 (80) | 142 (80) | 140 (79) |
| Ask the customer if they have their medication with them | 116 (65) | 122 (69) | 116 (65) |
| Suggest that the customer throw up | 74 (42) | 75 (42) | 78 (44) |
| A person with a food allergy can consume small quantities of the allergenic food without risk. | | | |
| Yes | 29 (16) | 40 (23) | 39 (22) |
| No (correct) | 87 (49) | 83 (47) | 78 (44) |
| Unsure or skipped | 62 (35) | 55 (31) | 61 (35) |
| A person with a food allergy can potentially experience a life-threatening reaction if they consume the food that they are allergic to. | | | |
| Yes (correct) | 40 (23) | 30 (17) | 31 (17) |
| No | 44 (25) | 56 (32) | 53 (30) |
| Unsure or skipped | 94 (53) | 92 (52) | 94 (53) |
| Removing a food allergen from the meal after it has been prepared is a method to ensure the meal is safe for a customer with food allergies. | | | |
| Yes | 30 (17) | 42 (24) | 45 (25) |

*(Continued)*

**Table 3.** (Continued)

| | Manager (N = 178) | Food worker (N = 178) | Server (N = 178) |
|---|---|---|---|
| No (correct) | 83 (47) | 70 (39) | 65 (37) |
| Unsure or skipped | 65 (36) | 66 (37) | 68 (38) |

(a) Responses are shown in the order in which they were asked. "n" denotes the number of managers and workers who answered the question affirmatively.

(mean = 3.81, SD = 0.634, n = 221, r = 0.04, P = 0.756), and 3.83 for servers (mean = 3.66, SD = 0.675, n = 197, r = 0.03, P = 0.452; Table 6).

As shown in Table 7, the total ANOVA model indicated significant group differences ($F_{2,626}$ = 4.96, P = 0.007). Post hoc tests revealed that workers (mean = 3.82, SD = 0.634, n = 221) and managers (mean = 3.86, SD = 0.636, n = 211) had significantly higher attitude scores than servers (mean = 3.66, SD = 0.675, n = 197). However, there was no statistically significant difference between food workers and staff managers.

## Multiple logistic regression analysis of manager, worker, and server attitudes

Regarding managers, multivariate logistic regression analysis revealed six factors as influential contributors to attitudes toward FA management (Table 8). Managers in restaurants catering to >10 meals for customers with FA in the preceding month were more likely to have positive attitudes toward FA than their counterparts in establishments serving ≤10 of such meals. Furthermore, managers intending to address queries from clients with FA had higher FA attitude scores. Managers in restaurants with designated personnel managing FA inquiries and requests had elevated FA attitude scores. Notably, managers in establishments incorporating allergen information into their menus had lower FA attitude scores than those in establishments without such information. Regarding experience, managers with a minimum of four years in the field were less likely to have higher FA attitude scores than those with less experience. Notably, managers who had received FA training at their establishments were more likely to nurture positive FA attitudes than those without such training.

**Table 4. Comparisons of food allergy knowledge and attitude scores according to groups.**

| Group | Mean difference | 95% confidence interval |
|---|---|---|
| **Knowledge scores[a]** | | |
| Manager vs. food worker | -0.514 | (-1.76, 0.732) |
| Manager vs. server | -0.246 | (−1.534, -1.042) |
| Server vs. food worker | -0.268 | (-1.541, 1.006) |
| **Attitude scores[b]** | | |
| Manager vs. food worker | 0.046 | (−0.077, 0.168) |
| Manager vs. server | 0.194 | (0.068, 0.320) * |
| Server vs. food worker | -0.148 | (−0.273, -0.024) * |

[a] one-way ANOVA ($F_{2,623}$ = 0.328, P = 0.720).

[b] one-way ANOVA ($F_{2,626}$ = 4.965, P = 0.007).

* Significant at P < 0.05.

A comprehensive ANOVA model showed no important differences in knowledge scores among the three groups ($F_{2,623}$ = 0.328, P = 0.720). Conversely, a statistically significant difference in attitude scores was observed among these groups ($F_{2,626}$ = 4.965, P = 0.007).

**Table 5. Results of the multiple logistic regression analysis of characteristics associated with restaurant managers, food workers, and servers scoring in the top 50% of food allergy knowledge scores.**

| Characteristics | OR (90% CI) | P |
|---|---|---|
| **Managers who scored in the top 50%** | | |
| The number of meals provided to customers with food allergies during the previous month | | 0.003 |
| 1–10 vs. 0 | 1.36 (0.75–3.57) | 0.208 |
| >10 vs. 1–10 | 4.21 (2.48–5.12) | 0.051 |
| >10 vs. 0 | 4.35 (2.78–5.63) | 0.001 |
| Specific person to contact with issues and concerns regarding food allergies | | |
| Yes vs. no | 1.53 (1.04–2.61) | 0.063 |
| **Food workers who scored in the top 50%** | | |
| Received food allergy training while employed at the restaurant | | |
| Yes vs. no | 4.32 (2.10–8.21) | <0.001 |
| Sex | | |
| Women vs. men | 2.36 (1.18–7.36) | 0.003 |
| Experience as a restaurant food worker | | |
| ≥2 vs. <2 years | 3.51 (1.34–4.27) | 0.005 |
| The most expensive food item listed on the menu | | |
| 51–100 SR vs. < 50 SR | 1.27 (1.30–5.46) | 0.022 |
| >100 SR vs. 51–100 SR | 0.67 (0.23–1.24) | 0.389 |
| >100 SR vs. <50 SR | 1.48 (0.81–4.23) | 0.228 |
| **Servers who scored in the top 50%** | | |
| Specific person to contact with issues and concerns regarding food allergies | | |
| Yes vs. no | 2.49 (1.33–4.66) | 0.016 |
| Service type | | |
| Full vs. quick service | 2.71 (1.40–5.24) | 0.014 |
| Number of meals served in a day | | |
| 101–300 vs. 1–100 | 1.02 (0.50–2.06) | 0.935 |
| >300 vs. 101–300 | 2.53 (1.21–5.37) | 0.024 |
| >300 vs. 1–100 | 2.61 (1.18–5.68) | 0.035 |

Note: The models were created using a forward selection criterion of $P < 0.10$. The variables are presented in the order of the steps in which they were included in the model.

OR, odds ratio; CI, confidence interval.

OR > 1 indicates that the odds of the outcome (knowledge score in the top 50%) were greater for the first-mentioned category (1 to 10) than for the second-mentioned category (0).

b$\chi^2$ = 17.18, df = 3, P < 0.001, N = 262.

c$\chi^2$ = 30.50, df = 5, P < 0.001, N = 192.

d$\chi^2$ = 16.97, df = 4, P = 0.002, N = 149.

**Table 6. Descriptive data of groups' attitude scores.**

| | Server | Worker | Manager |
|---|---|---|---|
| **Range, min–max** | 2.0–5.0 | 1.83–5.0 | 2.0–5.0 |
| **Mean ± SD** | 3.66 ± 0.675 | 3.81 ± 0.634 | 3.86 ± 0.636 |
| **Median (IQR)** | 3.60 (0.75) | 3.83 (0.67) | 4.0 (0.67) |

**Table 7. Comparison of attitude scores among the three groups.**

|            | Sum of Squares | df  | Mean Square | F     | P-value |
|------------|----------------|-----|-------------|-------|---------|
| Intergroup | 4.170          | 2   | 2.085       | 4.965 | 0.007   |
| Intragroup | 262.910        | 626 | 0.420       |       |         |
| Total      | 267.080        | 628 |             |       |         |

Regarding food workers, multivariate logistic regression analysis indicated four compelling factors associated with attitudes toward FA. Food workers in restaurants with an established strategy for addressing queries from patrons with FA were more likely to cultivate positive attitudes toward FA than those in establishments without such a plan. Furthermore, food workers with at least some college educations were more likely to have positive attitudes toward FA than their less-educated peers. Food workers in restaurants with fewer than five workers per manager were more likely to have higher FA attitude scores than counterparts in restaurants with five or more workers per manager. Furthermore, workers in chain restaurants were more likely to have positive attitudes toward FA than those in smaller establishments.

Regarding servers, multivariate logistic regression analysis revealed four distinctive factors intricately associated with attitudes toward FA. Servers with at least some college education were significantly more likely to nurture positive attitudes toward FA than their less-educated peers. In particular, servers who had received FA training at their restaurants were more likely to have positive attitudes toward FA than those without such training. In addition, servers in establishments equipped with strategies to address queries from patrons with FA had elevated FA attitude scores compared with those in establishments without such provisions. Finally, servers with a minimum of two years of experience in the restaurant industry were more likely to have positive attitudes toward FA than those with less experience.

## Discussion

To our knowledge, the present study is the first to evaluate FA knowledge, attitudes, and practices in restaurants in the Qassim region, Saudi Arabia. While some restaurant staff demonstrated knowledge of common allergens like peanuts, dairy products, and eggs, the study revealed a significant gap in their understanding of FA management and allergen handling practices. This includes a lack of awareness regarding the severity of reactions, proper first aid procedures, and cross-contamination prevention. These shortcomings pose a potential risk for customers with FA.

The study identified two critical knowledge gaps among restaurant staff. The first relates to the management of severe allergic reactions. While nearly all staff recognized the importance of calling emergency services, a significant portion lacked understanding of specific first aid measures, such as the ineffectiveness of providing water to customers with breathing difficulties. This highlights the need for comprehensive FA training that goes beyond emergency response to include proper first aid procedures.

The second knowledge gap concerns allergen handling practices. Despite regulations mandating allergen disclosure, the study found a conspicuous absence of allergen information in most restaurant menus and dining areas. Additionally, staff knowledge about the top eight allergens was suboptimal. This highlights the need for stricter enforcement of allergen disclosure regulations and enhanced training for staff on allergen identification and cross-contamination prevention.

Concerning workers' knowledge of first aid, we found that nearly all staff knew the vital role they play in food allergy response. For instance, they know to call 937 when a customer

**Table 8. Multiple logistic regression analysis of characteristics associated with restaurant managers, food workers, and servers scoring in the top 50% of food allergy attitude scores (a).**

| Characteristic | OR (90% CI) | P |
|---|---|---|
| **Managers who scored in the top 50%[b]** | | |
| Number of meals provided to customers with food allergies during the previous month | | <0.001 |
| 1–10 vs. 0 | 1.28 (0.72–2.28) | 0.567 |
| >10 vs. 1–10 | 3.71 (2.10–6.52) | 0.001 |
| >10 vs. 0 | 4.70 (2.34–9.76) | <0.001 |
| Restaurant strategy for responding to inquiries from customers with food allergies | | |
| Yes vs. no | 2.57 (1.58–4.51) | 0.002 |
| Specific person to contact with issues and concerns regarding food allergies | | |
| Yes vs. no | 1.61 (1.22–2.84) | 0.076 |
| Allergen information on the menu | | |
| Yes vs. no | 0.32 (0.21–0.59) | 0.012 |
| Experience as a restaurant manager | | |
| ≥4 vs. <4 years | 0.57 (0.34–0.92) | 0.072 |
| Received food allergy training while employed at the restaurant | | |
| Yes vs. no | 1.81 (1.20–2.82) | 0.085 |
| **Food workers who scored in the top 50%[c]** | | |
| Restaurant strategy for responding to inquiries from customers with food allergies | | |
| Yes vs. no | 2.53 (1.23–4.41) | 0.023 |
| Highest educational attainment | | |
| College level or higher vs. High school diploma or below | 3.25 (1.82–6.34) | 0.001 |
| Worker: manager ratio | | |
| <5:1 vs. ≥5:1 | 2.24 (1.27–4.34) | 0.021 |
| Restaurant type | | |
| Chain vs. independent | 2.14 (1.12–3.20) | 0.037 |
| **Servers who scored in the top 50%[d]** | | |
| Highest educational attainment | | |
| College level or higher vs. High school diploma or below | 3.23 (1.81–6.15) | 0.001 |
| Received food allergy training while employed at the restaurant | | |
| Yes vs. no | 2.70 (1.42–5.18) | 0.032 |
| Restaurant strategy for responding to inquiries from customers with food allergies | | |
| Yes vs. no | 2.53 (1.26–5.14) | 0.041 |
| Experience as a restaurant server | | |
| ≥2 vs. <2 years | 1.69 (1.21–3.82) | 0.072 |

The models were created using a forward selection criterion of P < 0.10. The variables are presented in the order in which they are included in the model.

Note: OR, odds ratio; CI, confidence interval.

OR > 1 indicates that the odds of the outcome (attitude score in the top 50%) were greater for the first-mentioned category (1–10) than for the second-mentioned category (0).

b $\chi^2$ = 52.00, df = 7, P < 0.001, N = 248.

c $\chi^2$ = 27.86, df = 4, P < 0.001, N = 196.

d $\chi^2$ = 24.43, df = 4, P < 0.001, N = 149.

experiences a severe allergic reaction, such as trouble breathing, which in most cases could be lifesaving. Therefore, most workers and servers understand the importance of emergency services in the event of an allergic reaction. Almost half of the staff thought drinking water could help customers with bad FA reactions, such as trouble breathing. In another study, Choi et al.

also found deficiencies in the response to emergency situations; for instance, most respondents were not knowledgeable about the use of injectable epinephrine in cases involving severe FAs [18].

The study conducted by Radke et al. (2017) revealed that approximately half of the reported fatal food allergy reactions over 13 years were caused by food from a restaurant or other food-service establishment [12]. Another study, conducted by Muñoz-Furlong and Weiss (2009), estimated that roughly 150 to 200 deaths occur annually due to FAs with a significant portion of these incidents occurring in settings such as restaurants [19]. These results emphasize the importance of proper FA management in restaurants. However, the results in this study show a lack of FA knowledge among restaurant staff. For instance, most respondents were not knowledgeable about the top eight food allergens from a given list of allergens [18]. Moreover, restaurant food handlers had no training and poor knowledge of the severity of FAs, and the importance of reading the list of ingredients shown on the label, as well as avoiding cross-contamination during the preparation of meals [20].

By contrast, managers showed higher knowledge of food allergens and allergic reaction symptoms than workers and servers. For example, a study conducted by Sogut et al. showed that almost all of the managers regularly read food labels to check for any ingredient that might cause a food allergy, while among food handlers the rate was 63% [21]. In this study, a multivariate logistic regression analysis revealed two factors associated with heightened FA knowledge scores among restaurant managers. First, a positive correlation was found between the frequency of meals served to clients with FAs in the prior month and higher FA knowledge scores, with managers who served more than 10 such meals demonstrating enhanced FA awareness. Second, the presence at an establishment of a specialized individual addressing FA inquiries and requests correlated with higher FA knowledge scores among managers at such establishments. These findings indicate that augmenting the frequency of meals served to patrons with FAs and designating specialized personnel to handle FA concerns may contribute to heightened FA knowledge and awareness among restaurant managers. However, a knowledge gap persists regarding the gravity of FAs and the appropriate handling of allergenic ingredients. For instance, almost half of the staff (51% of managers, 54% of food workers, and 57% of servers) thought that someone with a food allergy could safely eat small amounts of the food to which they are allergic. In their study, Radke et al. showed that more than 10% of managers and staff believed that someone with a food allergy could safely consume a small amount of that allergen [16]. These results highlight the necessity of providing a training program administered by the SFDA.

A noteworthy finding of this study is the conspicuous absence of allergen information in most restaurant menus. Instances where allergens were mentioned on menus or within dining areas were scarce. For example, only 14% of establishments listed allergens on their menus or food lists; even fewer possessed documentation or information panels catering to customers. In exceptional cases when documentation did exist, allergenic ingredients were predominantly denoted in Arabic and customers were requested to inform staff about allergies when ordering. This is despite SFDA mandating compliance with the Saudi Technical Regulation "SFDA.FD 56/2018," titled "Disclosure of allergens in the list of meals for food establishments that provide food to the consumer outside the home in 2019." Similarly, despite the EU-wide legislation introduced in December 2014 requiring restaurant servers to provide written and verbal information about one or more of the fourteen most common food allergens in their food, some studies have found that food allergens are not listed on menus or on other documents, such as on restaurant signs [22]. These results are concerning because people with FAs frequently rely on written allergen information to avoid possible allergens when dining out [23,24]. This emphasizes the importance of adhering to the policies of the SFDA and providing workers

with intensive training on allergens in the menu. Moreover, this gap underscores the urgent need for enhanced educational and training programs focused on the gravity of FAs and the meticulous handling of allergens [25].

Restaurant mode of operation was also associated with differences in FA attitude scores, with the post hoc analysis demonstrating that employees from corporate-owned restaurants had significantly more positive food allergy attitudes compared to employees from franchised establishments [24]. Food handlers in the chain restaurants were more prepared to meet the needs of customers with FAs than those in the independent restaurants [26]. This suggests that corporate policies and training programs may play a role in shaping staff attitudes and practices related to FAs. Our study found that only 24% of employees had received FA training during their tenure at their current restaurants, while 86% of the workers had served meals to individuals with FAs; this highlights a significant discrepancy between the availability of training and everyday practices, underscoring the urgent need to enhance educational and training programs focused on FAs to ensure customer safety and effective emergency response. These findings are comparable to a study conducted by Loerbroks et al., indicating that restaurants have limited knowledge, and customers cannot solely rely on their knowledge [27].

This synthesis of our study findings with preceding research highlights a consistent theme: there is a critical need to bridge the knowledge gap in FA awareness and management within the restaurant industry. Addressing this need through targeted educational initiatives could significantly mitigate the risks faced by individuals with FAs.

Our study had some limitations. First, we used a self-administered questionnaire to collect data, which may have been subject to social desirability and recall bias. Second, our survey consisted of a multiple-choice format, indicating that participants could be restricted to choosing the most appropriate answer rather than their actual answers. Third, we represented a region rather than a country in this study. Consequently, we propose the need for future research designs encompassing a broader geographical scope and larger sample sizes to enrich our study's insights.

Despite these limitations, our study provides valuable insights into the knowledge gap in FA awareness and management within the restaurant industry in the Qassim region, Saudi Arabia. The findings underscore the urgent need for targeted educational initiatives and stricter enforcement of regulations to bridge this gap and ensure the safety of customers with FA.

## Conclusions

FA safety measures in restaurants are imperative, and there is a pressing need for well-defined policies and pertinent guidance regarding FAs. The level of knowledge among restaurant personnel varies considerably. Restaurant staff members often lack formal FA training, depending instead on their own understanding and personal experiences. Notably, disparities in information might arise depending on specific restaurants within a region, which may be influenced by factors such as the prevalence or reach of restaurant chains. Therefore, research centered on the restaurant milieu could improve the caliber of employee training and ultimately mitigate anaphylactic events and avert potential fatalities.

## Acknowledgments

We thank the Environmental Health Specialists Network (EHS-Net) for designing and publishing the protocol for studying food allergies in restaurants for the study that was conducted by the Friends of Patients Association in Unaizah, based on all the study protocols. We also acknowledge the hardworking members of the data collection team (Tala Abdullah Almutlaq, Tala Mansour Almousa, Raghad Saleh Alhabib, Lojain Ahmed Algraawi, and Yara Saleh

Almuqrin) who were crucial to the accomplishment of this study. Their commitment and efforts are greatly appreciated. Finally, we thank the administrative and technical assistance provided to conduct this study successfully.

## Institutional review board statement

The study was conducted in accordance with the Declaration of Helsinki and approved by the Institutional Ethics Committee of the Qassim Region (registration number 607-44-5119, approval date: October 31, 2022).

## Informed consent statement

Informed consent was obtained from all participants involved in the study.

## Author Contributions

**Conceptualization:** Ghadi A. Alkhalaf, Norah A. Aljuaylan.

**Data curation:** Ghadi A. Alkhalaf, Norah A. Aljuaylan, Jolan S. Alsaud.

**Formal analysis:** Ghadi A. Alkhalaf, Norah A. Aljuaylan, Jolan S. Alsaud, Fatima R. Aljalaood.

**Investigation:** Ghadi A. Alkhalaf, Norah A. Aljuaylan, Fatima R. Aljalaood.

**Methodology:** Ghadi A. Alkhalaf, Norah A. Aljuaylan, Jolan S. Alsaud.

**Resources:** Ghadi A. Alkhalaf, Norah A. Aljuaylan, Boshra A. Aljokaidb, Fatima R. Aljalaood.

**Software:** Ghadi A. Alkhalaf, Norah A. Aljuaylan.

**Validation:** Ghadi A. Alkhalaf, Norah A. Aljuaylan, Jolan S. Alsaud.

**Writing – original draft:** Ghadi A. Alkhalaf, Norah A. Aljuaylan, Jolan S. Alsaud, Boshra A. Aljokaidb, Fatima R. Aljalaood.

**Writing – review & editing:** Jolan S. Alsaud.

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
