## [Decision Letter · Decision Letter 0]

11 Mar 2024

PONE-D-23-40795Restaurant staff’s knowledge, practices, and attitudes pertaining to food allergy in Qassim region, Saudi Arabia: A cross-sectional analysisPLOS ONE

Dear Dr. Aljuaylan,

Thank you for submitting your manuscript to PLOS ONE. After careful consideration, we feel that it has merit but does not fully meet PLOS ONE’s publication criteria as it currently stands. Therefore, we invite you to submit a revised version of the manuscript that addresses the points raised during the review process.

We look forward to receiving your revised manuscript.

Kind regards,

Ali B. Mahmoud, Ph.D.

Academic Editor

PLOS ONE

Journal Requirements:

3. In the ethics statement in the Methods, you have specified that verbal consent was obtained. Please provide additional details regarding how this consent was documented and witnessed, and state whether this was approved by the IRB.

Reviewers' comments:

Reviewer's Responses to Questions

**Comments to the Author**

1. Is the manuscript technically sound, and do the data support the conclusions?

Reviewer #1: Yes

Reviewer #2: Yes

2. Has the statistical analysis been performed appropriately and rigorously? 

Reviewer #1: Yes

Reviewer #2: Yes

3. Have the authors made all data underlying the findings in their manuscript fully available?

Reviewer #1: Yes

Reviewer #2: Yes

4. Is the manuscript presented in an intelligible fashion and written in standard English?

Reviewer #1: Yes

Reviewer #2: Yes

5. Review Comments to the Author

Reviewer #1: Food allergy work is important and relevant as the number of people with food allergies continues to increase.

The article has good information but the discussion of results is limited. The authors need to add more references and comprehensively discuss the results, and clearly identify limitations, future research.

Reviewer #2: The manuscript presented in an intelligible fashion and written in standard Language. I have the following queries or suggestions: Firstly, The authors should identify what is meant by '' scored in the top 50%". On what bases you choose 50%?

Secondly, In Multiple logistic regression analysis of manager, worker, and 318

server attitudes (Table 8): the authors used the same variables of knowledge as the same as table 5. Attitudes are commonly measured using Likert scaling, which involves using a scale with multiple equivalent items to produce a summated score. Your responses are in knowledge domains not in attitude domain. Please take this point and clarify the exact domains of the attitude you used.

6. PLOS authors have the option to publish the peer review history of their article (what does this mean?). If published, this will include your full peer review and any attached files.

Reviewer #1: No

Reviewer #2: **Yes: **Moawia Gameraddin

---

## [Author Response · Author response to Decision Letter 0]

29 Jun 2024

Based on the reviewers comment #5 " The authors need to add more references and comprehensively discuss the results, and clearly identify limitations, future research.", the results and discussion has been revised. We have answered the question asked by reviwer2 below:

The question:

The manuscript presented in an intelligible fashion and written in standard Language. I have the following queries or suggestions: Firstly, the authors should identify what is meant by " scored in the top 50%". On what bases you choose 50%?

Our answer:

For the knowledge score, we summed the number of correct answers (out of 19) and used each group’s median score to dichotomize the participants as having more or less knowledge.

For the attitude score, we assigned point values to each response as follows: strongly disagree = 1, disagree = 2, unsure = 3, agree = 4, and strongly agree = 5. We then averaged each participant’s response to the five attitude questions. We used each group’s median score to divide participants into those having relatively positive or less positive attitudes. This has been added to the methodology section.

Line (394-89) and Table 1 We changed Food service to Foodservice.

Line 35: We changed (omitted) to (did not post)

Line 36: We changed (varies) to (varied)

Line 38: We changed (food safety) to (food allergy measures)

Line 59: We add sesame to the list of allergens.

Line 151: Remove the word (we) 

Additionally, Table 6 has been added as it was observed that overall knowledge was not measured. This is very important to determine the need for additional training courses.

About this point (Table 1: I suggest you divide the demographic data in more detail. Example: Non-Asian restaurants can be of so many cuisines) A good suggestion, but unfortunately, we cannot make the adjustment now as the data was collected based on the original classification.

Response to the Ethical Considerations Regarding Obtaining the Consent:

Thank you for your inquiry regarding obtaining the consent. A waiver of documentation of informed consent is requested in accordance with 45 CFR 46.117(c). The proposed research meets the first criterion for the waiver, as the probability and magnitude of harm or discomfort in participation are not greater in and of themselves than those ordinarily encountered in daily life. Additionally, as the

research involves no procedures for which written consent is normally required outside the research context, the study also meets the second criterion for waiver. Before conducting the study, we obtained verbal informed consent from the restaurant managers. 

The member read for the manager a short introduction describing the purpose of the

study and how the data will be used. Then, the interviewer asked the manager if he or she agrees to participate in the study. The interview proceeded for those who agreed to participate. The interviewer read a brief recruiting and informed consent script to the worker/server who was identified as a potential participant by the manager. We requested a waiver of written informed consent; thus, neither managers nor workers/servers read the informed consent scripts.

Kind regards,

Norah Aljuaylan

---

## [Decision Letter · Decision Letter 1]

7 Aug 2024

PONE-D-23-40795R1Restaurant staff’s knowledge, practices, and attitudes pertaining to food allergy in Qassim region, Saudi Arabia: A cross-sectional analysisPLOS ONE

Dear Dr. Aljuaylan,

Thank you for submitting your manuscript to PLOS ONE. After careful consideration, we feel that it has merit but does not fully meet PLOS ONE’s publication criteria as it currently stands. Therefore, we invite you to submit a revised version of the manuscript that addresses the points raised during the review process.

We look forward to receiving your revised manuscript.

Kind regards,

Ali B. Mahmoud, Ph.D.

Academic Editor

PLOS ONE

Journal Requirements:

Reviewers' comments:

Reviewer's Responses to Questions

**Comments to the Author**

1. If the authors have adequately addressed your comments raised in a previous round of review and you feel that this manuscript is now acceptable for publication, you may indicate that here to bypass the “Comments to the Author” section, enter your conflict of interest statement in the “Confidential to Editor” section, and submit your "Accept" recommendation.

Reviewer #1: All comments have been addressed

Reviewer #2: All comments have been addressed

2. Is the manuscript technically sound, and do the data support the conclusions?

Reviewer #1: Yes

Reviewer #2: Yes

3. Has the statistical analysis been performed appropriately and rigorously? 

Reviewer #1: Yes

Reviewer #2: Yes

4. Have the authors made all data underlying the findings in their manuscript fully available?

Reviewer #1: Yes

Reviewer #2: Yes

5. Is the manuscript presented in an intelligible fashion and written in standard English?

Reviewer #1: Yes

Reviewer #2: Yes

6. Review Comments to the Author

Reviewer #1: The revisions are done well. I strongly suggest the authors proof-read the article carefully to correct any issues with writing and grammar.

Reviewer #2: I have read the manuscript, all sections are well revised except the abstract and methodology section.

In abstract section, the authors should add the main findings rather than other general statements such as (Among the 178 restaurants included in the analysis, 97% of the managers and servers were men, and most had not completed food allergy courses. Notably, 86% of the participants omitted any form of food allergy information from their operations.) this statement should removed and put another main findings instead of it.

In methodology section, I suggest to divide the methodology section into subdivisions so as to be easily identified and read.

7. PLOS authors have the option to publish the peer review history of their article (what does this mean?). If published, this will include your full peer review and any attached files.

Reviewer #1: **Yes: **Anirudh Naig

Reviewer #2: No

---

## [Author Response · Author response to Decision Letter 1]

28 Aug 2024

We thank the reviewers for their valuable comments, which have significantly contributed to the enhancement of our research. In response to your recommendations:

Improvement of the Abstract: We have revised the abstract to more clearly and accurately reflect the main findings of the study. General statements have been removed and replaced with specific results that highlight the significance and extent of our findings.

Restructuring the Methodology Section: The methodology section has been divided into several subsections to facilitate reading and understanding. Each subsection addresses a specific aspect of the methodology, assisting the reader in following the steps we have taken in an organized and detailed manner.

Based on the reviewers comment #5 " The authors need to add more references and comprehensively discuss the results, and clearly identify limitations, future research.", the results and discussion has been revised. We have answered the question asked by reviwer2 below:

The question:

The manuscript presented in an intelligible fashion and written in standard Language. I have the following queries or suggestions: Firstly, the authors should identify what is meant by " scored in the top 50%". On what bases you choose 50%?

Our answer:

For the knowledge score, we summed the number of correct answers (out of 19) and used each group’s median score to dichotomize the participants as having more or less knowledge.

For the attitude score, we assigned point values to each response as follows: strongly disagree = 1, disagree = 2, unsure = 3, agree = 4, and strongly agree = 5. We then averaged each participant’s response to the five attitude questions. We used each group’s median score to divide participants into those having relatively positive or less positive attitudes. This has been added to the methodology section.

Line (394-89) and Table 1 We changed Food service to Foodservice.

Line 35: We changed (omitted) to (did not post)

Line 36: We changed (varies) to (varied)

Line 38: We changed (food safety) to (food allergy measures)

Line 59: We add sesame to the list of allergens.

Line 151: Remove the word (we) 

Additionally, Table 6 has been added as it was observed that overall knowledge was not measured. This is very important to determine the need for additional training courses.

About this point (Table 1: I suggest you divide the demographic data in more detail. Example: Non-Asian restaurants can be of so many cuisines) A good suggestion, but unfortunately, we cannot make the adjustment now as the data was collected based on the original classification.

Response to the Ethical Considerations Regarding Obtaining the Consent:

Thank you for your inquiry regarding obtaining the consent. A waiver of documentation of informed consent is requested in accordance with 45 CFR 46.117(c). The proposed research meets the first criterion for the waiver, as the probability and magnitude of harm or discomfort in participation are not greater in and of themselves than those ordinarily encountered in daily life. Additionally, as the

research involves no procedures for which written consent is normally required outside the research context, the study also meets the second criterion for waiver. Before conducting the study, we obtained verbal informed consent from the restaurant managers. 

The member read for the manager a short introduction describing the purpose of the

study and how the data will be used. Then, the interviewer asked the manager if he or she agrees to participate in the study. The interview proceeded for those who agreed to participate. The interviewer read a brief recruiting and informed consent script to the worker/server who was identified as a potential participant by the manager. We requested a waiver of written informed consent; thus, neither managers nor workers/servers read the informed consent scripts.

Kind regards,

Norah Aljuaylan

---

## [Decision Letter · Decision Letter 2]

24 Sep 2024

Restaurant staff’s knowledge, practices, and attitudes pertaining to food allergy in Qassim region, Saudi Arabia: A cross-sectional analysis

PONE-D-23-40795R2

Dear Dr. Aljuaylan,

We’re pleased to inform you that your manuscript has been judged scientifically suitable for publication and will be formally accepted for publication once it meets all outstanding technical requirements.

Kind regards,

Ali B. Mahmoud, Ph.D.

Academic Editor

PLOS ONE

Additional Editor Comments (optional):

Reviewers' comments:

Reviewer's Responses to Questions

**Comments to the Author**

1. If the authors have adequately addressed your comments raised in a previous round of review and you feel that this manuscript is now acceptable for publication, you may indicate that here to bypass the “Comments to the Author” section, enter your conflict of interest statement in the “Confidential to Editor” section, and submit your "Accept" recommendation.

Reviewer #1: All comments have been addressed

2. Is the manuscript technically sound, and do the data support the conclusions?

Reviewer #1: Yes

3. Has the statistical analysis been performed appropriately and rigorously? 

Reviewer #1: Yes

4. Have the authors made all data underlying the findings in their manuscript fully available?

Reviewer #1: Yes

5. Is the manuscript presented in an intelligible fashion and written in standard English?

Reviewer #1: Yes

6. Review Comments to the Author

Reviewer #1: (No Response)

7. PLOS authors have the option to publish the peer review history of their article (what does this mean?). If published, this will include your full peer review and any attached files.

Reviewer #1: **Yes: **Anirudh Naig

---

## [Editor Report · Acceptance letter]

27 Sep 2024

PONE-D-23-40795R2 

PLOS ONE

Dear Dr. Aljuaylan, 

I'm pleased to inform you that your manuscript has been deemed suitable for publication in PLOS ONE. Congratulations! Your manuscript is now being handed over to our production team.

Kind regards, 

on behalf of

Dr. Ali B. Mahmoud 

Academic Editor

PLOS ONE